# Prone Positioning Decreases Inhomogeneity and Improves Dorsal Compliance in Invasively Ventilated Spontaneously Breathing COVID-19 Patients—A Study Using Electrical Impedance Tomography

**DOI:** 10.3390/diagnostics12102281

**Published:** 2022-09-21

**Authors:** Charalampos Pierrakos, Fleur L. I. M. van der Ven, Marry R. Smit, Laura A. Hagens, Frederique Paulus, Marcus J. Schultz, Lieuwe D. J. Bos

**Affiliations:** 1Department of Intensive Care & Laboratory of Experimental Intensive Care and Anesthesiology (LEICA), Amsterdam University Medical Centers, Location ‘AMC’, 1105 AZ Amsterdam, The Netherlands; 2Department of Intensive Care, Brugmann University Hospital, Université Libre de Bruxelles, 1020 Bruxelles, Belgium; 3Mahidol–Oxford Tropical Medicine Research Unit (MORU), Mahidol University, Bangkok 73170, Thailand; 4Nuffield Department of Medicine, University of Oxford, Oxford OX1 2JD, UK

**Keywords:** acute respiratory distress syndrome (ARDS), mechanical ventilation, intensive care images, recruitment, alveolar distention, global inhomogeneity index, silent spaces, end-expiratory lung impedance

## Abstract

Background: We studied prone positioning effects on lung aeration in spontaneously breathing invasively ventilated patients with coronavirus disease 2019 (COVID-19). Methods: changes in lung aeration were studied prospectively by electrical impedance tomography (EIT) from before to after placing the patient prone, and back to supine. Mixed effect models with a random intercept and only fixed effects were used to evaluate changes in lung aeration. Results: fifteen spontaneously breathing invasively ventilated patients were enrolled, and remained prone for a median of 19 [17 to 21] hours. At 16 h the global inhomogeneity index was lower. At 2 h, there were neither changes in dorsal nor in ventral compliance; after 16 h, only dorsal compliance (βFe +18.9 [95% Confidence interval (CI): 9.1 to 28.8]) and dorsal end-expiratory lung impedance (EELI) were increased (βFe, +252 [95% CI: 13 to 496]); at 2 and 16 h, dorsal silent spaces was unchanged (βFe, –4.6 [95% CI: –12.3 to +3.2]). The observed changes induced by prone positioning disappeared after turning patients back to supine. Conclusions: in this cohort of spontaneously breathing invasively ventilated COVID-19 patients, prone positioning decreased inhomogeneity, increased lung volumes, and improved dorsal compliance.

## 1. Introduction

Prone positioning can improve oxygenation in acute respiratory distress syndrome (ARDS) patients with severe hypoxemia [1,2,3,4]. Prone positioning is widely used in patients with ARDS related to coronavirus disease 2019 (COVID-19) [5]. Nevertheless, the mechanism of beneficial effects of prone position in COVID-19 has not been thoroughly investigated. Oxygenation changes may not track lung mechanics alterations during the prone position [6]. Electrical impedance tomography is a non-invasive method that can be used for the assessment of lung aeration and local mechanics in patients with ARDS related to COVID-19 [7].

Most investigations that focused on changes in lung aeration induced by prone positioning have been performed in deeply sedated and paralyzed patients. In our experience, most of these patients could be placed in the prone position without receiving neuromuscular blocking agents. In fact, we did hardly use extra sedation in these patients, and consequently, these patients remained spontaneously breathing while in the prone position [8].

We assessed the effects of prone positioning on global and local changes in lung aeration in critically ill COVID-19 patients that were placed in the prone position while spontaneously breathing. This adds to our current understanding, which is mainly driven by observations of deeply sedated and paralyzed patients. We hypothesized that prone positioning would decrease the inhomogeneity of aeration and recruit collapsed lung tissue. We, therefore, assessed several aeration parameters by electric impendence tomography (EIT), early and late after placing patients in the prone position, and also determined the effects of placing a subject back into the supine position. 

## 2. Materials and Methods

### 2.1. Design and Setting

This was a single-center prospective observational study in invasively ventilated COVID-19 patients admitted to the intensive care unit (ICU) in the Amsterdam University Medical Centers, locations ‘AMC’, Amsterdam, The Netherlands, during the second wave of the national outbreak. The Local Ethics Committee approved the study (W20_545#20.605), and informed consent was waived as in our ICU both prone positioning and EIT monitoring were part of the standard care.

### 2.2. Inclusion and Exclusion Criteria

Patients were eligible if (1) COVID-19 pneumonia was proven by means of RT–PCR for the SARS-CoV-2; (2) they were receiving invasive ventilation with an assisted ventilation mode, and (3) were planned for the prone position because of refractory hypoxemia. Exclusion criteria were the presence of a pacemaker or one or more thoracic tubes. For practical reasons we also excluded patients that were contaminated or had an infection with one or more multi-resistant bacteria, and when there was no EIT monitor available.

All patients were treated according to local protocols. Muscular relaxant treatment was avoided, and a shift to assisted ventilation mode was made as soon as possible after invasive ventilation initiation. PEEP was titrated up to 12 cm H_2_O permitting a V_T_ 6−8 mL per predicted body weight (PBW). When V_T_ was higher than 8 ml/PBW, pressure support was decreased to a minimum of 5 cm H_2_O. Prone position was applied when the P_a_O_2_/FiO_2_ ratio was lower or equal to 150 mmHg.

### 2.3. Data Collected

Baseline demographics were captured for each subject. The extent and density of alveolar opacities on chest X-rays within the last day before the prone positioning session were scored using the ‘Radiographic Assessment of Lung Edema’ (RALE) score [9]. Dead space was calculated using the Enghoff modification of the Bohr equation. Dynamic global compliance was calculated by dividing V_T_ by the difference between maximum total respiratory system pressure and positive end-expiratory pressure (PEEP).

EIT images were collected 1 hour before initiation of prone positioning, 2 and 16 h after initiation of prone positioning, and 2 h after return to the supine position. Vital signs, ventilator parameters, and blood gas analysis results were collected 1 hour before prone positioning, at 2 and 16 h after initiation of prone positioning, and after 2 h in the supine position.

### 2.4. Electrical Impedance Tomography

A silicone electrical impedance tomography belt with 16 electrodes connected to a PulmoVista^®^ 500 EIT monitor (Dräger Medical GmbH, Lübeck, Germany) was placed around the chest at the 4th intercostal space, and a reference electrode was placed in the abdominal area. The belt remained attached during the whole prone positioning session, and thereafter at 2 h in the supine position, without any further adjustments.

EIT images were recorded at 40 Hz. Images were digitally filtered using a low–pass filter with a cutoff of 40/min. EIT images were analyzed in line using the EITdiag software^®^ (Dräger Medical GmbH). The following parameters were calculated and presented according to the previously suggested order [10]:
(A)Ventilation map:
Local dynamic compliance—calculated by dividing the local proportion of total tidal impedance variation calibrated to V_T_ by the difference between maximum total respiratory system pressure and PEEP;End-expiratory lung impedance changes (dEELI)—average lower impedance of value of the electrical impedance variation of each respiratory cycle was measured to define EELI. An increase in the EELI is considered as an increase in the end-expiratory global and regional lung volumes. EELI before the prone position was used as the reference value to calculate dEELI [11]; andSilent spaces—defined as the number of pixels with minimal impedance variation less than 10% to maximal impedance and a correlation coefficient > 40% of the maximum linear correlation coefficient with the global impedance waveform [12].(B)EIT-based indices:
Center of ventilation—synonym center of gravity calculated by dividing the ventilated area into equal space slices. The sum of ventilation-related impedance change of each slice is calculated and presented as a bar and is plotted on dorsal-to-ventral thorax diameter. The horizontal line position, which divides in equal impedance changes, is expressed in percent of dorsal-to-ventral thorax diameter––a percentage > 50% represents more ventral ventilation distribution [13,14];Global inhomogeneity index (GI)—calculated as the ratio of the sum of absolute differences between the median value of tidal variation and every single pixel value to the sum of all impedance values––a higher GI means more inhomogeneous distribution of the VT [15]; and(C)Ventilatory desynchrony:
The presence of pendelluft and breath stacking phenomenon were evaluated visually.

### 2.5. Study Endpoints

The main endpoints were the global homogeneity index, ventral and dorsal local compliance, EELI, and silent spaces.

### 2.6. Analysis Plan

Demographic, clinical, and outcome variables were presented as percentages for categorical variables and as medians with interquartile ranges (IQR) for continuous variables.

Dorsal and ventral regions of interest (ROIs) were symmetrical and were defined by dividing lung images. We used mixed effect models with a random intercept and only fixed effects analysis using the R package *lme4* to quantify the changes in aeration during and after prone positioning. Herein, measurements before initiation of prone positioning were used as a reference for all successive measurements, i.e., at 2 and 16 h after initiation of prone positioning and at 2 h after the subject was placed back in the supine position. A random intercept was used for each subject to account for variation in the baseline values. Fixed effects were estimated and reported with 95% confidence intervals (CI). Recruitment of lung tissue was considered to occur if compliance, EELI and silent spaces was improved.

All analyses were performed in R through the R-studio interface (R v3.3.1. https://cran.r-project.org/bin/windows/base/ (accessed on 10 April 2022)). A *p*–value < 0.05 was considered significant.

## 3. Results

### 3.1. Patients and Prone Positioning

We included 15 spontaneously breathing invasively ventilated COVID-19 patients, all with severe ARDS. Baseline characteristics and respiratory parameters before prone positioning are presented in Table 1. Patients were invasively ventilated with assisted ventilation. Ventilation distribution was predominately dorsal with the median center of ventilation at 47.8 [46.1 to 51.2] %. Prone positioning was initiated median 4 [3 to 6] days after the start of ventilation. In one subject, prone positioning was stopped at 3 h because of a technical problem with the central venous line. In the other patients, the duration of the first prone positioning session lasted a median of 19 [17 to 21] hours. Propofol alone was used for patients’ anesthesia. Propofol dose was not increased during the prone position with a maximum dose of 3.3 [2.8 to 3.5] mg/kg/h (βFe + 0.06 [95% CI: −0.07 to 0.19]) (Table 2). The level of PEEP (PEEP) remained unchanged during the prone position with a maximum level of PEEP at 16 h prone position 11 [10 to 12] cm H_2_O (βFe −4.8 [95% CI: −11.3 to 1.12]). No major modifications in pressure support were observed during the prone position with a maximum level at 16 h prone position of 7 (5−13) cm H_2_O (βFe +0.78 [95% CI: –1.06 to 2.64]). The prone position did not increase significantly the PaO_2_/FiO_2_ (βFe +7.41 [95% CI: –4.06 to 19.3]) and did not significantly decrease dead space (βFe −4.89 [95% CI: –11.01 to 1.28]) (Figure 1 and Appendix A).

EIT belt remained attached to the subject during the prone position and 2 h after the end of the prone position without causing any trauma or irritation. Figure 2 shows representative images obtained with EIT during and after the prone position.

### 3.2. Prone Positioning-Induced EIT Changes

(A)Ventilation map (Figure 3 and Appendix A):
Local dynamic compliances were unchanged at 2 h after initiation of prone positioning. At 16 h after initiation of prone position dorsal compliance increased whereas ventral compliance was unchanged;EELI increased only in ventral areas at 2 h after initiation of prone positioning. At 16 h after initiation of prone position either the dorsal and ventral EELI increased; andSilent spaces increased in ventral areas and were unchanged at 2 h after initiation of the prone position. At 16 h after initiation of prone position, silent spaces in either ventral and dorsal areas were unchanged.(B)EIT-based indices (Figure 1 and Appendix A):
Center of ventilation remained unchanged during the prone position; andGlobal inhomogeneity index was unchanged at 2 h after initiation of prone position but decreased at 16 h.(C)Ventilatory desynchrony (Table 2):
No changes in the incidence of pendelluft phenomenon and breath stacking.

### 3.3. Supine Positioning-Induced EIT Changes

(A)Ventilation map (Figure 3 and Appendix A):
Local dynamic compliances returned to baseline when patients were placed back into the supine position;EELI in either dorsal and ventral areas returned to baseline when patients were placed back into the supine position; andSilent spaces were unchanged in either ventral and dorsal areas when patients were placed back into the supine position.(B)EIT-based indices: (Figure 1 and Appendix A)
Center of ventilation remained unchanged compared to baseline values;Global inhomogeneity index returned to baseline values when patients were placed back.(C)Ventilatory desynchrony (Table 2):
No changes in the incidence of pendelluft phenomenon and breath stacking.

### 3.4. Lung Recruitment

Lung recruitment (i.e., the concomitant increase in local compliance, EELI and decrease in local silent spaces) at 2 h after initiation of prone position was observed for ventral areas in 1 subject (6%) and for dorsal areas in 4 patients (26%). At 16 h after initiation of prone position lung recruitment in ventral areas was observed in 2 patients (13%) and in dorsal areas in 8 patients (53%). When patients were placed back into the supine position, lung recruitment in ventral areas was observed in none of the patients and in dorsal areas in 6 patients (40%). One subject had at the same time recruitment of ventral and dorsal areas at 16 h after initiation of prone position.

## 4. Discussion

The findings of this study in spontaneously breathing invasively ventilated COVID-19 patients with severe ARDS can be summarized as follows: (1) prone position decreased ventilation inhomogeneity as it was indicated by decreasing global inhomogeneity index; (2) dynamic compliance and EELI increased in dorsal areas but silent spaces did not decrease showing a local improvement in aeration without evidence of lung tissue recruitment; and (3) all EIT variables recede to baseline values early after return to the supine position.

COVID-19 pneumonia is characterized by a mixed morphology of consolidations and rounded-shaped ground glass opacities [16,17]. Importantly, these lesions present a diffuse patchy distribution and can affect anterior and posterior lung areas [17]. As thus, the loss of aeration is not dominant in the dependent lung areas [18]. Patients with this lung morphology could benefit less from prone positioning [19]. During prone positioning, we observed an improvement in the compliance of dorsal areas associated with an increase in the EELV, which implies that the aeration of dorsal areas improved. Compliance decreased in ventral areas despite the increase in EELV, and overdistention of ventral areas is the most likely explanation for this finding [20]. Thus, our results suggest that prone positioning may cause an increase in the transpulmonary pressure which can increase aeration, particularly in dorsal areas but also overdistention of other lung areas.

Extrapolating the pressure–volume loop of the global lung for lung areas, it could be argued that improvement in the local compliance could be sufficient to define lung recruitment in these areas [21,22]. Nevertheless, we did not observe any decrease in the silent segments. A likely explanation of our findings is that the prone position improved chest wall compliance which resulted in an increase in transpulmonary pressure and shifting of several lung units from ventilation under inflection point to their maximal compliance ventilation. Of note, previous reports have demonstrated a paradoxical increase in respiratory system compliance after chest or abdominal compression in patients with COVID-19 ARDS [23,24,25]. Therefore, the results of our study imply that improved local compliance is related more to lung aeration improvement rather than to lung recruitment.

In our study, patients did not receive any muscle paralysis, but the sedation level target was high. As such, an elevated dose of propofol was given as the sole sedative agent which potentially may affect the activity of the diaphragm [26]. However, the administrated dose was much lower than 6 mg/kg/h which was found previously to affect diaphragmatic function in a clinical setting [27] and close to or even lower than the dosages reported in previous studies when a combination of propofol and midazolam was used [28,29]. Furthermore, one previous study showed a significant respiratory drive in invasively ventilated patients with a Richmond Agitation scale of −5 [30]. We observed that ventilation was predominant in the dorsal areas before proning. This finding is different from previous studies in patients with ARDS due to another cause wherein patients received muscle paralysis [31,32,33,34]. Additionally, several patients in our cohort presented the pendelluft phenomenon. Therefore, the majority of the patients in this study had an active and functioning diaphragm despite the given sedation.

Predominant dorsal ventilation before prone positioning could negatively moderate the effects of prone positioning [31,32]. However, we observed improved aeration and compliance of the dorsal areas. Given that we evaluated changes in aeration in patients who did not receive neuromuscular blocking agents, it can be speculated that the observed improvements were related to the patient’s respiratory effort. Notwithstanding, prone position in invasively ventilated patients is associated with less respiratory effort [35] and less diaphragmatic function [36], which would decrease apparent compliance and not improve it. We also observed a decrease in the global inhomogeneity index, which is opposite to the effect expected after an increase in the respiratory effort [37], and no increase in the pendelluft phenomenon incidence. Hence, in this small cohort of invasively ventilated patients with diaphragmatic activity, the observed increase in compliance during the prone position is unlikely to represent an increase in the patient’s respiratory effort. Improved lung aeration and ventilation homogeneity due to the prone position is the most plausible alternative explication.

We kept the EIT belt attached to patients before, during, and after prone position, so the measurements were comparable to baseline values. Our experience shows that this is feasible without causing any thoracic trauma. This strategy in combination with the absence of other interventions during the study period permitted to assess the effects of the prone position on lung function, differently from previous studies which used EIT to assess PEEP titration in the prone position. A weakness of our approach was that the chronological distance between the measurements was long and we cannot exclude that factors other than prone position, e.g., fluid accumulation and suction events, could affect our measurements. Furthermore, despite using suggested filters to eliminate artifacts and high electrode signal quality being verified before each measurement, we cannot exclude an interaction between belt performance and thorax wall conduct during prone position. Nevertheless, all these factors could result in a decrease in end-expiratory impedance and an increase in tidal variation of impedance in ventral areas. Additionally, the impedance variation and EELI after the end of the prone position were comparable to baseline values. Based on these considerations, technical issues stemming from patients’ position changes and the long duration of measurements can only have biased the results towards the null and would not influence our conclusions.

The main strength of this study was the systematic evaluation of the aeration changes in spontaneously breathing invasively ventilated patients. Early spontaneously breathing is increasingly used, and the benefits of deep sedation and muscle paralysis are increasingly questioned [38], also in COVID-19 patients [39,40]. We investigated prone positioning–induced temporal changes in lung aeration longitudinally, including before and after the episode patients were in a prone position. In our study, prone positioning was the standard of care, and the team was very well used to accepting spontaneous breathing in these patients.

There are also limitations. The sample size was small, albeit comparable to previous studies [31,33,34]. Changes in aeration, however, were compared within each subject. The patients enrolled in this cohort probably had notable respiratory activity as is indicated by low-pressure support and the existence of the pendelluft phenomenon. However, inspiratory and expiratory pressures generated by the patients’ efforts were not evaluated since esophagus catheters are not standardly used in our unit. We did not include deeply sedated and paralyzed patients. The findings of our study may not be translatable to patients ventilated with a controlled ventilation mode. For lung recruitment assessment was used as one of the variable changes in silent segments which has been suggested as an easily applicable method to monitor recruitment and overdistention [12,41,42]. Nevertheless, monitoring changes in silent segments during prone position has not been evaluated thoroughly and has not been compared with computed tomography findings. Finally, the time between initiation of ventilation and the start of this study differed between patients, because prone positioning was only applied in case refractory hypoxemia developed.

In our cohort, we did neither observe significant changes in PaO_2_/FiO_2_ nor in dead space. In previous studies, the response of PaO_2_/FiO_2_ to prone positioning had no associations with outcomes [43,44]. Prone positioning could confer benefits mainly by preventing or minimizing ventilator-associated lung injury. Our findings fit into this idea. The improved homogeneity and compliance are indicative of better, or more lung-protective mechanical ventilation. However, we also observed an increase in silent spaces in ventral lung parts which may explain the absence of a decrease in dead space during prone position which is associated with lower mortality [45]. Therefore, our data suggest that despite the beneficial global effects on lung function prone position may be harmful to some lung areas due to overdistention. EIT can be used to identify these effects of prone position at the bedside in patients with spontaneous breathing. Whether the local overdistention related to prone position has an impact on the patient’s outcome and EIT-guided adjustment of ventilatory parameters aiming to alleviate these effects can have palpable benefits for the patients with ARDS should be further investigated in future studies.

## 5. Conclusions

Electrical impedance tomography can be used in clinical practice for monitoring prone position-induced changes in aeration in spontaneously breathing patients. In this small cohort of spontaneously breathing invasively ventilated COVID-19 patients with severe ARDS, prone positioning resulted in decreased inhomogeneity, increased aeration, and improved dorsal compliance despite no significant changes in oxygenation. The effects of the prone position were largely reversed after placing patients back into a supine position. Longer durations of prone positioning, however, should be seen in the context of the possibility of failure of recruitment of collapsed lung tissue and the possibility of overdistention of anterior lung areas.

## Figures and Tables

**Figure 1 diagnostics-12-02281-f001:**
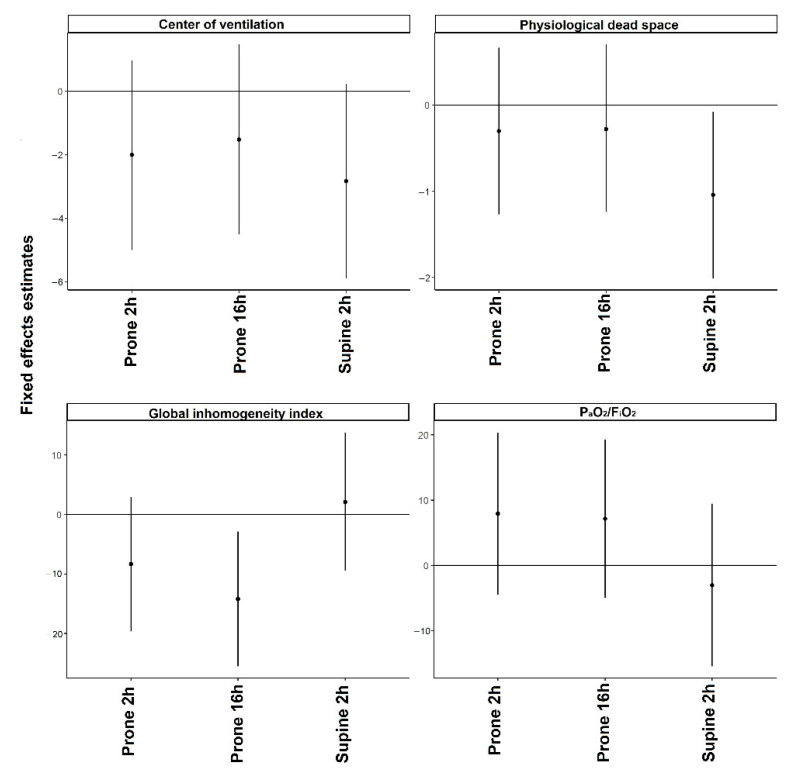
Mixed effect models with a random intercept and only fixed effects using as a fixed variable the examination time and dependent variables the global ventilatory parameters. Values before prone position (PP) were considered reference values. The regression coefficients and 95% confidence intervals indicate the probability of each variable to be higher or lower compared to the reference value at each time point.

**Figure 2 diagnostics-12-02281-f002:**
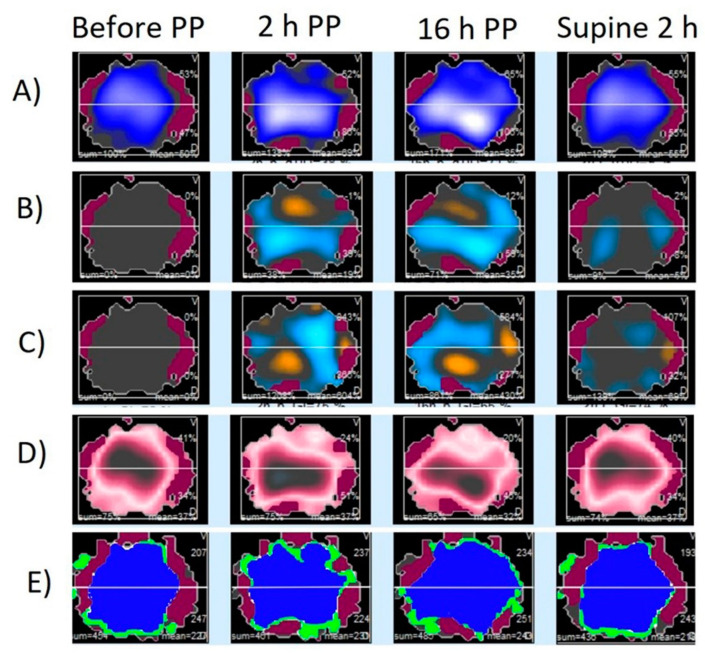
Regional electrical impedance derived parameters during the different study phases in a representative patient. Panel (**A**): Impedance map; Panel (**B**): Impedance changes distribution compared to values before prone position (PP), increases are presented with blue color and decreases with orange; Panel (**C**): End-expiratory impedance changes distribution compared to values before PP, increases are presented with blue color and decreases with orange; Panel (**D**): Global inhomogeneity index map. The bigger the difference between the tidal variation and the median tidal, the darker the color turns. Panel (**E**): Map of well-ventilated pixels (blue color), poorly ventilated pixels (green color), and not ventilated pixels (red color).

**Figure 3 diagnostics-12-02281-f003:**
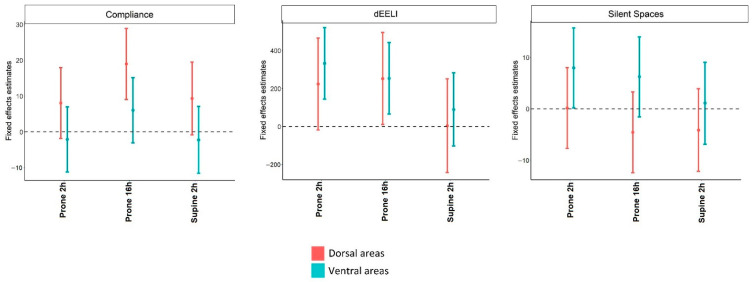
Mixed effect models with a random intercept and only fixed effects using as fixed variable, the examination time, and dependent variables, local compliance, changes in end-expiratory lung impedance, and silent spaces. Values before PP were considered reference values. The regression coefficients and 95% confidence intervals indicate the probability of each variable to be higher or lower compared to the reference value at each time point.

**Table 1 diagnostics-12-02281-t001:** Patients’ characteristics.

Variables	
Number of patients	15
Age, years (IQR)	62 (56–71)
Sex, female (%)	7 (46)
Charlson Comorbidity Index (IQR)	5 (3–7)
Time after intubation, days (IQR)	4 (3–6)
Duration of prone position, hours (IQR)	19 (17–21)
Tidal volume, mL (IQR)	438 (395–512)
Tidal volume, mL/kg PBW (IQR)	7.6 (5.7–9.7)
PEEP, cmH_2_O (IQR)	10 (9–13)
PaO_2_/FiO_2_, mmHg (IQR)	82 (54–115)
spO_2_/FiO_2_ (IQR)	118 (94–153)
Compliance, mL/cmH_2_O (IQR) *	45 (30–59)
Mechanical power, J/min (IQR) *	22.9 (13.6–26.3)
RALE score (IQR)	25 (23–35)

**PBW**: Per predictive body weight, PEEP: Positive end-expiratory pressure, **PaO_2_**: Partial arterial pressure of oxygen, **FiO_2_:** Fraction of inspired oxygen, **spO_2_**: Saturation of peripheral oxygen, **RALE:** Radiographic Assessment of Lung Edema. *: Calculated under assisted ventilation mode.

**Table 2 diagnostics-12-02281-t002:** Hemodynamic and respiratory parameters during the different phases of the study.

	Supine	2 h Prone	16 h Prone	2 h Supine
**Hemodynamic parameters**				
Heart Rate, beats/min (IQR)	99 (83–102)	92 (83–112)	94 (89–119)	95 (77–119)
Systolic Arterial Pressure, mmHg (IQR)	115 (108–138)	123 (112–136)	119 (102–129)	118 (112–122)
Diastolic Arterial Pressure, mmHg (IQR)	68 (66–71)	62 (60–70)	63 (59–68)	52 (51–63) *
Mean Arterial Pressure, mmHg (IQR)	90 (82–92)	82 (77–88)	80 (74–88)	74 (72–81) *
**Respiratory parameters**				
Respiratory Rate, breaths/min (IQR)	31 (20–35)	23 (18–33)	22 (16–31)	19 (13–25)
Pressure Support, cmH_2_O (IQR)	5 (5–12)	6 (5–12)	7 (5–13)	8 (5–13)
PEEP, cmH_2_O (IQR)	10 (9–13)	10 (10–12)	11 (10–12)	10 (10–12)
Tidal Volume, mL/kg (IQR)	7.1 (5.5–9.2)	7.1 (6.4–8.7)	8.3 (6.7–9.9)	7.7 (6.7–10.4) *
SpO_2_/FiO_2_, (IQR)	118 (94–153)	122 (107–137)	135 (105–159) *	118 (101–148)
Compliance mL/cmH_2_O (IQR)	45 (30–59)	45 (26–68)	72 (29–93) *	53 (34–71)
Breath Stacking, *n* (%)	5 (33)	5 (33)	4 (28)	3 (23)
Pendelluft Phenomenon, *n* (%)	7 (46)	4 (26)	5 (35)	2 (15)
**Sedation**				
Propofol, mg/kg/h (IQR)	3.1 (2.6–4.1)	3.1 (2.6–4.1)	3.3 (2.8–3.5)	3.1 (2.6–3.8)
RASS, (IQR)	−5 (−5–−4)	−5 (−5–−4)	−5 (−5–−4)	−5 (−5–−4)

**SpO_2_**: Saturation of peripheral oxygen, **FiO_2_**: Fraction of inspired oxygen, **RASS**: Richmond Agitation-Sedation Scale, *: Statistical significant change compared to baseline.

## Data Availability

The datasets generated during this study are available from the corresponding author on reasonable request due to privacy restrictions.

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
