# Peer review of "Prone Positioning Decreases Inhomogeneity and Improves Dorsal Compliance in Invasively Ventilated Spontaneously Breathing COVID-19 Patients—A Study Using Electrical Impedance Tomography"

_diagnostics, 2022, doi:10.3390/diagnostics12102281_

Round 1

Reviewer 1 Report

1)Abstract: We studied prone positioning effects on lung aeration in spontaneously breathing invasively ventilated patients with coronavirus disease 2019 (COVID–19). Changes in lung aeration were  studied prospectively by electrical impedance tomography (EIT) from before to after placing patient  prone, and back to supine. Linear mixed–effects models were used to evaluate changes in lung aer-ation. Fifteen spontaneously breathing invasively ventilated patients were enrolled, remained prone  for median of 19 [17 to 21] hours. At 16 hours the global inhomogeneity index was lower. At 2 hours, there were neither changes in dorsal nor in ventral compliance; after 16 hours, only dorsal compli-ance (βFE +18.9 [95%–CI 9.1 to 28.8]) and dorsal end expiratory lung impedance (EELI) was in-  creased (βFE, +252 [95%–CI 13 to 496]); at 2 and 16 hours, dorsal ‘silent spaces’ was unchanged (βFE, –4.6 [95%–CI –12.3 to +3.2]). The observed changes induced by prone positioning disappeared after  turning patients back to supine. In this cohort of spontaneously breathing invasively ventilated  COVID–19 patients, prone positioning decreased inhomogeneity, increased lung volumes, and improved dorsal compliance. The abstract is quite rumbling and difficult to read. Please, divide it in several section (for example background, aim, conclusions, …).

2) 1. Introduction L30-35. Prone positioning can improve oxygenation in acute respiratory distress syndrome  (ARDS) patients with severe hypoxemia [1–4]. Most investigations that focused on  changes in lung aeration induced by prone positioning have been performed in deeply  sedated and paralyzed patients. Prone positioning is widely used in patients with ARDS related to coronavirus disease 2019 (COVID–19) [5]. Please improve this section regarding COVID-19 and add these references:

A Bedside Selection of Positive End Expiratory Pressure by Electrical Impedance Tomography in Patients Undergoing Veno-Venous Extracorporeal Membrane Oxygenation Support: A Comparison between COVID-19 ARDS and ARDS from Other Etiologies. Journal of Clinical Medicine. 2022; 11(6):1639. https://doi.org/10.3390/jcm11061639

B Different Methods to Improve the Monitoring of Noninvasive Respiratory Support of Patients with Severe Pneumonia/ARDS Due to COVID-19: An Update. J Clin Med. 2022 Mar 19;11(6):1704. doi: 10.3390/jcm11061704. 

3) 1. Introduction L 40-45. We assessed the effects of prone positioning on global and local changes in lung aer-ation in critically ill COVID–19 patients that were placed in the prone position while spon-taneously breathing. We hypothesized that prone positioning would decrease the inho-  mogeneity of aeration and recruit collapsed lung tissue. We therefore assessed several aeration parameters by electric impendence tomography (EIT), early and late after placing  patients in the prone position, and also determined the effects of placing a subject back  into the supine position. Please ameliorate the study aim and underline the novelty of the study.

4) 3. Results 136 3.1. Patients and prone positioning. Please underline the most important statistical values to support the data.

5) 4. Discussion L216-220. The findings of this study in spontaneously breathing invasively ventilated COVID–19 patients with severe ARDS provide evidence that prone position decreases ventilation inhomogeneity without causing evident recruitment of lung tissue, and that aeration of  dorsal lung areas improves progressively during prone position but recede to baseline values early after return to the supine position. Please summarise and underline the most important statistical significant results of the study.

6) 4.Discussion L222. COVID–19 ARDS is characterized by a diffuse loss of aeration that is not dominant  in the dependent lung areas [14]. Please, improve these paragraph and add these references:

A DW-UNet: Loss Balance under Local-Patch for 3D Infection Segmentation from COVID-19 CT Images. Diagnostics 202111, 1942. https://doi.org/10.3390/diagnostics11111942

B The role of chest CT in deciphering interstitial lung involvement: systemic sclerosis versus COVID-19. Rheumatology (Oxford). 2022 Apr 11;61(4):1600-1609. doi: 10.1093/rheumatology/keab615.

7) 5. Conclusions L322-326 In this cohort of spontaneously breathing invasively ventilated COVID–19 patients  with severe ARDS, prone positioning resulted in decreased inhomogeneity, increased aeration, and improved dorsal compliance. Prone positioning, however, did not result in recruitment of collapsed lung tissue, and the effects of were largely reversed after placing  patients back into a supine position.

Please improve the description of conclusions and underline the clinical implication of the study.

Author Response

Reviewer 1:

  1. Abstract: Please, divide it in several section (for example background, aim, conclusions, …).

According to the journal guides for authors ‘abstract should be a single paragraph and should follow the style of structured abstracts, but without headings’. We added before its structure ‘background’, ‘methods’ , ‘results’ and ‘conclusions’.

‘Abstract: Background: We studied prone positioning effects on lung aeration in spontaneously breathing invasively ventilated patients with coronavirus disease 2019 (COVID–19). Methods: Changes in lung aeration were studied prospectively by electrical impedance tomography (EIT) from before to after placing patient prone, and back to supine. Mixed effect models with a random intercept and only fixed effects used to evaluate changes in lung aeration. Results: Fifteen spontaneously breathing invasively ventilated patients were enrolled, remained prone for median of 19 [17 to 21] hours. At 16 hours the global inhomogeneity index was lower. At 2 hours, there were neither changes in dorsal nor in ventral compliance; after 16 hours, only dorsal compliance (βFE +18.9 [95%–CI 9.1 to 28.8]) and dorsal end expiratory lung impedance (EELI) was increased (βFE, +252 [95%–CI 13 to 496]); at 2 and 16 hours, dorsal ‘silent spaces’ was unchanged (βFE, –4.6 [95%–CI –12.3 to +3.2]). The observed changes induced by prone positioning disappeared after turning patients back to supine. Conclusions: In this cohort of spontaneously breathing invasively ventilated COVID–19 patients, prone positioning decreased inhomogeneity, increased lung volumes, and improved dorsal compliance.’ (Page 1;line 14)

  1. Introduction L30-35. ‘Prone positioning can improve oxygenation in acute respiratory distress syndrome  (ARDS) patients with severe hypoxemia [1–4]. Most investigations that focused on  changes in lung aeration induced by prone positioning have been performed in deeply  sedated and paralyzed patients. Prone positioning is widely used in patients with ARDS related to coronavirus disease 2019 (COVID–19) [5].’ Please improve this section regarding COVID-19 and add these references:A) Bedside Selection of Positive End Expiratory Pressure by Electrical Impedance Tomography in Patients Undergoing Veno-Venous Extracorporeal Membrane Oxygenation Support: A Comparison between COVID-19 ARDS and ARDS from Other Etiologies. Journal of Clinical Medicine. 2022; 11(6):1639. https://doi.org/10.3390/jcm11061639 and B) Different Methods to Improve the Monitoring of Noninvasive Respiratory Support of Patients with Severe Pneumonia/ARDS Due to COVID-19: An Update. J Clin Med. 2022 Mar 19;11(6):1704. doi: 10.3390/jcm11061704. 

Thank you for this suggestion. We improved this paragraph accordingly:

‘Prone positioning can improve oxygenation in acute respiratory distress syndrome (ARDS) patients with severe hypoxemia [1–4]. Prone positioning is widely used in patients with ARDS related to coronavirus disease 2019 (COVID–19) [5]. Nevertheless, the mechanism of beneficial effects of prone position in COVID-19 have not been thoroughly investigated .Oxygenation changes may not track lung mechanics alterations during prone position [6]. Electrical impedance tomography is a non-invasive method that can be used for the assessment of lung aeration and local mechanics in patients with ARDS related to COVID-19 [7].’  (Page 1;line 32)

  1. Introduction L 40-45. ‘We assessed the effects of prone positioning on global and local changes in lung aeration in critically ill COVID–19 patients that were placed in the prone position while spontaneously breathing. We hypothesized that prone positioning would decrease the inhomogeneity of aeration and recruit collapsed lung tissue. We therefore assessed several aeration parameters by electric impendence tomography (EIT), early and late after placing  patients in the prone position, and also determined the effects of placing a subject back  into the supine position’. Please ameliorate the study aim and underline the novelty of the study.

We improved the aim of the study trying to reveal the novelty of the study:

We assessed the effects of prone positioning on global and local changes in lung aeration in critically ill COVID–19 patients that were placed in the prone position while spontaneously breathing. This adds to our current understanding, which is mainly driven by observations in deeply sedated and paralyzed patients. We hypothesized that prone positioning would decrease the inhomogeneity of aeration and recruit collapsed lung tissue. We therefore assessed several aeration parameters by electric impendence tomography (EIT), early and late after placing  patients in the prone position, and also determined the effects of placing a subject back  into the supine position’ (Page 2;line 46)

  1. Results 136 3.1. Patients and prone positioning. Please underline the most important statistical values to support the data.

We added statistical values to support the presented in this paragraph:

‘3.1. Patients and prone positioning

We included 15 spontaneously breathing invasively ventilated COVID–19 patients, all with severe ARDS. Baseline characteristics and respiratory parameters before prone positioning are presented in Table 1. Patients were invasively ventilated with assisted ventilation. Ventilation distribution was predominately dorsal with the median center of ventilation at 47.8 [46.1 to 51.2] % . Prone positioning was initiated median 4 [3 to 6] days after start of ventilation. In one subject, prone positioning was stopped at 3 hours because of a technical problem with the central venous line. In the other patients, du-ration of the first prone positioning session lasted median 19 [17 to 21] hours. Propofol alone was used for patients anesthesia. Propofol dose was not increased during prone position with maximum dose of 3.3 [2.8 to 3.5] mg/kg/h (βFE +0.06 [95%–CI -0.07 to 0.19]) (Table 2).The level of PEEP (PEEP) remained unchanged during prone position with maximum level of PEEP at 16 h prone position 11 [10 to 12] cm H2O (βFE -4.8 [95%–CI -11.3 to 1.12]). No major modifications in pressure support were observed during prone position with maximum level at 16 h prone position of 7 (5−13) cm H2O (βFE +0.78 [95%–CI –1.06 to 2.64]). Prone position did not increase significantly the PaO2/FiO2 (βFE +7.41 [95%–CI –4.06 to 19.3]) and did not significantly decrease dead space (βFE -4.89 [95%–CI –11.01 to 1.28]) (Figure 1, Figure S1). (Page 4; line 145)

  1. Discussion L216-220. ‘The findings of this study in spontaneously breathing invasively ventilated COVID–19 patients with severe ARDS provide evidence that prone position decreases ventilation inhomogeneity without causing evident recruitment of lung tissue, and that aeration of  dorsal lung areas improves progressively during prone position but recede to baseline values early after return to the supine position.’ Please summarise and underline the most important statistical significant results of the study.

We rephrase this paragraph accordingly underline statistical significant results of the study:

‘The findings of this study in spontaneously breathing invasively ventilated COVID–19 patients with severe ARDS can be summarized as follows: 1) prone position decreased ventilation inhomogeneity as it was indicated by decreasing global inhomogeneity index; 2) dynamic compliance and EELI increased in dorsal areas but ‘silent spaces’ did not decrease showing a local improvement in aeration without evidence of lung tissue recruitment and 3) all EIT variables recede to baseline values early after return to the supine position.’ (Page 9;line 231)

  1. Discussion L222. COVID–19 ARDS is characterized by a diffuse loss of aeration that is not dominant  in the dependent lung areas [14]. Please, improve these paragraph and add these references: A)DW-UNet: Loss Balance under Local-Patch for 3D Infection Segmentation from COVID-19 CT Images. Diagnostics 202111, 1942. https://doi.org/10.3390/diagnostics11111942 and B) The role of chest CT in deciphering interstitial lung involvement: systemic sclerosis versus COVID-19. Rheumatology (Oxford). 2022 Apr 11;61(4):1600-1609. doi: 10.1093/rheumatology/keab615.

Thank you for this suggestion. We improved this paragraph as follows:

‘COVID- 19 pneumonia is characterized by a mixed morphology of consolidations and rounded shaped ground glass opacities[16,17]. Importantly, these lesions present a diffuse patchy distribution and can affect anterior and posterior lung areas [17]. As thus, the loss of aeration is not dominant in the dependent lung areas[18]. Patients with this lung morphology could benefit less from prone positioning [19]. During prone positioning, we observed an improvement in the compliance of dorsal areas associated with an increase in the EELV, which implies that the aeration of dorsal areas improved. Compliance decreased in ventral areas despite the increase in EELV, and overdistention of ventral areas is the most likely explanation for this finding [20]. Thus, our results suggest that prone positioning may cause an increase in the transpulmonary pressure which can increase aeration, particularly in dorsal areas but also overdistention of other lung areas.’ (Page 9;line  238)

  1. Conclusions L322-326 ‘In this cohort of spontaneously breathing invasively ventilated COVID–19 patients  with severe ARDS, prone positioning resulted in decreased inhomogeneity, increased aeration, and improved dorsal compliance. Prone positioning, however, did not result in recruitment of collapsed lung tissue, and the effects of were largely reversed after placing  patients back into a supine position.’ Please improve the description of conclusions and underline the clinical implication of the study.

We improved the description of conclusions emphasizing the clinical implications of the study:

Electrical impedance tomography can be used in clinical practice for monitoring prone position induced changes in aeration in spontaneously breathing patients. In this small cohort of spontaneously breathing invasively ventilated COVID–19 patients with severe ARDS, prone positioning resulted in decreased inhomogeneity, increased aeration, and improved dorsal compliance despite not significant changes in oxygenation. The effects of prone position were largely reversed after placing patients back into a supine position. Longer durations of prone positioning, however, should be seen in the context of the possibility of failure of recruitment of collapsed lung tissue and the possibility of over-distention of anterior lung areas. (Page 11;line 341)

Reviewer 2 Report

Sample size is way too small – only 15.

Statistical models mentioned differently in the earlier and later part of manuscript -  ‘Linear mixed–effects models’ vs ‘The fixed effect regression model’ – which one is the correct final one?

Patients’ comorbidity scores/conditions – like Charlson or Elixhauser index – should be reported.

In the limitation section, mentioned – “The patients enrolled in this cohort probably had a significant respiratory activity as it is indicated by low pressure support and the existence of ‘pendelluft’ phenomenon.” – when using the word of ‘significant’ – should also listed the statistical test and p-value to support the claim.

Incomplete sentence in ‘Conclusion’ section – “the effects of were largely…”

Author Response

Reviewer 2

  1. Sample size is way too small – only 15.

We agree with the reviewer that this is a limitation of this study. We acknowledge this limitation in the discussion and we emphasize this in the conclusions as follows:

‘The sample size was small, albeit comparable to previous studies [31, 33, 34].’ (Page:10;line:311)

In this small cohort of spontaneously breathing invasively ventilated COVID–19 patients with severe ARDS, prone positioning resulted in decreased inhomogeneity, increased aeration, and improved dorsal compliance despite not major changes in oxygenation.(Page11; line 342)

2. Statistical models mentioned differently in the earlier and later part of manuscript -‘Linear mixed–effects models’ vs ‘The fixed effect regression model’ – which one is the correct final one?

‘Mixed effect models with a random intercept and only fixed effects’ is changed everywhere in the text and in the figure legends.

  1. Patients’ comorbidity scores/conditions – like Charlson or Elixhauser index – should be reported.

Charlson Comorbidity Index (CCI) was added in the Table 1.

  1. In the limitation section, mentioned – “The patients enrolled in this cohort probably had a significant respiratory activity as it is indicated by low pressure support and the existence of ‘pendelluft’ phenomenon.” – when using the word of ‘significant’ – should also listed the statistical test and p-value to support the claim.

We replaced the word significant:

‘The patients enrolled in this cohort probably had a notable respiratory activity as it is indicated by low pressure support and the existence of ‘pendelluft’ phenomenon.’ (Page 10 ; line 313)

  1. Incomplete sentence in ‘Conclusion’ section – “the effects of were largely…”

Corrected